# An Unethical Organizational Behavior for the Sake of the Family: Perceived Risk of Job Insecurity, Family Motivation and Financial Pressures

**DOI:** 10.3390/ijerph19116541

**Published:** 2022-05-27

**Authors:** Ibrahim A. Elshaer, Marwa Ghanem, Alaa M. S. Azazz

**Affiliations:** 1Department of Management, College of Business Administration, King Faisal University, Al-Ahsaa 31982, Saudi Arabia; 2Hotel Studies Department, Faculty of Tourism and Hotels, Suez Canal University, Ismailia 41522, Egypt; 3Tourism Studies Department, Faculty of Tourism and Hotels, Suez Canal University, Ismailia 41522, Egypt; marwamagdy00@gmail.com or; 4Department of Tourism and Hospitality, Arts College, King Faisal University, Al-Ahsaa 31982, Saudi Arabia

**Keywords:** unethical organizational behaviors, job insecurity, family financial pressure, family motivation, COVID-19 pandemic, tourism

## Abstract

In organizations, unethical behaviors are pervasive and costly, and considerable recent research attention has been paid to various types of workplace unethical behavior. This study examines employees’ behaviors that are carried out for the benefit of one’s family but violate societal and organizational moral standards. Drawing upon the self-maintenance and bounded ethicality theories, this study examines the engagement of unethical organization behaviors (UOB) in the name of the family during the COVID-19 pandemic. It examines the influence of job instability and the mediating role of family financial pressure and family motivation. A total of 770 employees in hotels and travel agents in Egypt were targeted, and the data were analyzed using structural equation modeling. The results posit that perceived risk of job insecurity predicts engagement in unethical organizational behaviors, while intentions of UOB increase by high family motivation and financial pressures. Toward the end of this paper, a discussion on the theoretical and practical implications and are presented.

## 1. Introduction

Job loss and job insecurity were among the topics that were of most concern as consequences of the worldwide spread of the coronavirus. Export-dependent economies and economies that rely on tourism have struggled adjusting to fluctuating and shifting demand. The World Travel and Tourism Council (WTTC) suggested that the global job market was at risk for 75 million people in 2020 [1], while the World Economic Forum reported that the lockdown and layoff practices during the COVID-19 pandemic resulted in 114 million job losses in 2020 [2]. Even employees who survive the layoffs become anxious about their career and suffer high levels of job insecurity [3,4]. According to prior studies, job insecurity is a significant hindrance-related stress that negatively influence tourism business’s ability to achieve its desired work [4,5,6] and can lead to absenteeism and anxiety [7].

As a result of the COVID-19 pandemic and according to social cognitive theory, employees who face job insecurity in addition to increased family financial strain are more likely to use moral disengagement practices to disable moral self-regulation, resulting in increased levels of unethical behavior. Moreover, researchers asserted that the heavy losses suffered by business organizations created significant unethical organizational practice [3,8,9]. Approximately 90% of companies indicate that COVID-19 is a risk to ethical behavior at work, according to a report from Ernst and Young [10]. Similarly, in a survey conducted in India by Bhattacharyya [11] during the COVID-19 outbreak, many employees were found to be willing to engage in unethical behavior, such as falsifying records of customers (32%) and disclosing false information to their managers (29%).

Researchers have recently been interested in researching the reasons behind unethical practices during the COVID-19 pandemic and understanding the relationship with job loss and perceived job insecurity. For example, guided by appraisal theories of emotion, Hillebrandt and Barclay [4] argued that COVID-19 provokes anxiety and can drive employees to prioritize their self-interest and promote cheating behavior in workplace. Elshaer and Azazz [3] surveyed 650 employees working in the Egyptian tourism industry to explore the psychological process that would drive unethical organizational behaviors by employees who contend with job insecurity. They found that perceived job insecurity reduces job embeddedness, strengthens turnover intentions, and encourages unethical behavior.

In addition, previous studies asserted that employees who suffer from stresses due to workplace threats (e.g., job insecurity) may conduct UOB as a way to protect their gains and job assets [12,13]. Employees conducting unethical organization behavior (UOB) can also be driven by self-serving interest to acquire personal gains [14] or benefiting their organization or group [15] while benefiting themselves accordingly [16]. Based on behavioral ethics research, people can generally fail to make an objective assessment of the ethics of their behavior in the workplace [17], since their cognitive biases cause them to underestimate or ignore their unethical behavior. Elshaer et al. [18] added that often, employees do not make an explicit decision to act unethically but rather seek to convince themselves that there is nothing wrong with their behavior. In general, UOB can lead to devastating effects, such as significant financial losses, legal prosecutions, and corporate closures [19,20]. While even simple unethical behaviors in organizations can lead to significant hidden costs, tarnishing employee morale and damaging a company’s reputation [21].

Despite the thrive of behavioral ethics research, negative behavior displayed within organizations has such a wide scope that it is virtually not possible to explore within the scope of a few research projects [22], and various studied contexts are needed to unpack the drivers of UOB for mitigating resulting risks. Most previous research on unethical behavior in the workplace focus on unethical pro-organizational behavior (in the name of the company) [9] with little attention to unethical practices in the name of self-interest [22,23,24] or the family [2,25]. The prevalent unethical behaviors during the COVID-19 pandemic [3] and their possible relations with job insecurity [2] have raised significant questions about the different forms of unethical organizational behavior (UOB) during crises, the possible psychological process that drive such practices, and how it can be mitigated. Therefore, to address this gap of research and based on theories of conservation of resources, social cognitive and behavioral ethics (i.e., the self-maintenance and bounded ethicality theories), the current study aims to further investigate the effect of job insecurity on unethical organizational behavior among employees amid the COVID-19 pandemic, using family financial pressure and family motivation as mediating variables. The results of this study, thus, extend prior research results on conditions that shape unethical practices in the workplace and better explain the widespread UOB during the COVID-19 pandemic. It also provides insights into how organizations can address ethical challenges.

## 2. Theoretical Background and Hypotheses Development

### 2.1. Job Insecurity and Unethical Workplace Behavior

Job insecurity has long been a subject of study in a wide variety of research papers [2,4,26,27]. Numerous studies have been conducted in the hotel and tourism industries, notably on job insecurity and its effect on human behavior [2,28]. Job insecurity is a “perceptual phenomenon” that focuses on a person’s current job stability threats [29]. Hellgren et al. [30] proposes two categories of job insecurity: quantitative “threats to the job as a whole” and qualitative “threats to desired job characteristics”. Quantitative job insecurity focuses on the expected job loss triggered by intentional or unintentional administrative signals or appraisal reports by employees’ supervisors, while qualitative job insecurity illustrates how an individual perceives their future job loss in light of a perceived threat [30].

Given the devastating effects of the COVID-19 pandemic on the economy, downsizing has become a common strategy in recent years. Downsizing is a method of reducing labor costs, streamlining operations, and increasing organizational competitiveness [31]. According to [32,33], organizational restructuring and the downsizing process have proved to threaten workers and their careers, resulting in exacerbating perceived job insecurity [34]. The resulting stresses of perceived job instability may motivate employees to engage in unethical actions that they believe might protect them against the threat of job loss or even keep some important features of their job [23,35,36,37,38,39]. Unethical workplace behavior may include actions that benefit the organization, group or employee self-interest, such as diminishing colleagues’ efforts to improve personal relationships, reputations, and professional success [40]. Employees’ activities and behaviors that are in direct conflict with the organization’s norms and values may create significant financial losses [41] and jeopardize organizational image [42]. Accordingly, as shown in Figure 1, the below hypothesis is suggested:

**Hypothesis** **1** **(H1).***Job insecurity has a positive impact on workplace unethical behavior*.

### 2.2. Job Insecurity, Family Financial Pressure, and Unethical Workplace Behavior

Employees as well as their families face financial difficulties as a result of the high percentage of job insecurity [43]. Despite remarkable advances in our understanding of the impact of job insecurity on well-being, stress, and health over the last several years [30,44], it is difficult to infer causality. Financial stress on the family (i.e., related to satisfying basic needs, family education cost, utilities payments, or family healthcare expenses) is likely to exacerbate job insecurity, which in turn leads to financial pressure [45,46]. A few research studies have explored the relationship between job insecurity and employees’ financial well-being and pressure and provide contradicting results between significant [47] and insignificant [45] effects. Therefore, the relationship between job insecurity and family financial pressure requires further investigation.

Families’ financial difficulties are not just felt by the impoverished; they are also felt by rich people who want to maintain pace with their friends. When confronted with strong financial difficulties from family members, an employee’s principal purpose is to relieve those pressures. The more pressing the need, the more significant this goal will become. Generally, supporting one’s family monetarily is a key worth in human culture [48]. Liu et al. [25] elaborated that social expectations are framed, and laws are laid out to uphold the satisfaction of familial monetary obligations. When stressed to assist their families, employees are bound to consider it to be their responsibility to make any strides important to help their family, subsequently obscuring their moral obligation regarding their actions.

Based on the theory of conservation of resources developed by Hobfoll [12], when people face a concern of losing their valuable resources, they become pressured to protect those resources by, for example, acquiring recuperation assets. Accordingly, when employees encounter substantial financial difficulties in their families, they are more likely to concentrate their efforts on obtaining financial compensation from their employer [46]. As a result, self-justification of immoral actions in the workplace can then thrive [49]. Unethical workplace activities may help alleviate the stress and aggravation felt by employees while simultaneously improving the financial well-being of the employees’ families. Many sorts of unethical behavior in the name of the family are directly linked to financial advantages that might relieve financial stress, such as bringing organization possessions home for use or accompanying relatives to the workplace to gain benefit from the organization’s resources.

According to the social cognitive theory proposed by Bandura [50] and the self-concept maintenance theory developed by Mazar et al. [51], the readiness of self-justifications encourages unethical behavior through an expanded moral disengagement. Self-justifications can make the UB looks less immoral; costs of the dishonest action are limited, disregarded, or confounded; or casualties of the wrongdoing are undervalued or accused. In summary, when family financial pressures are increased because of perceived job insecurity, employees may become more likely to participate in unethical workplace behavior to benefit their family and decrease these related stresses. Thus, the following hypotheses were proposed:

**Hypothesis** **2** **(H2).***Job insecurity has a significant impact on family financial pressure*.

**Hypothesis** **3** **(H3).***Family financial pressure has a significant impact on unethical workplace behavior*.

**Hypothesis** **4** **(H4).***Family financial pressure mediates the impact of job insecurity and unethical workplace behavior*.

### 2.3. Job Insecurity, Family Motivation, and Unethical Workplace Behavior

Supporting one’s family is a significant justification for why many people work, yet surprisingly few researchers have investigated the effects of family motivation [47], particularly in relation to perceived pressures and job insecurity. Moreover, Liu et al. [25] noted that no previous study has thoroughly examined the role of the family as a motivating factor for unethical behavior in organizations. Menges et al. ([47], p. 700) defines family motivation as *“the desire to expend effort to benefit one’s family”.* Prior research suggested that workers with a high family motive are more likely to prioritize family concerns and perceive the family’s best interests as a main consideration [25,46,48]. This would likely drive employees who place a high value on their families to justify immoral behavior in the workplace since it benefits them personally and socially as well as their own families [52,53].

Relatedly, previous studies [53,54,55] explained that employees ted to perceive their desire protect to benefit another party’s or beneficiaries’ interest (family members in this study) as a moral (ease-to-use) justification for unethical behavior. Based on the concept of bounded ethicality, an employee may often behave unethically as he/she either consciously or unconsciously was able to disregard and justify his/her own misconduct [30].

Accordingly, this study suggests that when job insecurity increases, workers with high family motivation may engage in unethical practices to benefit their family interests and will demonstrate less attention to organizational moral standards. In other words, whether individuals choose to be engaged in unethical organizational behavior (UOB) that violates the moral code and interests of the firm can be determined by their familial motives. Employees thus become more prone to justify their unethical workplace behavior in order to alleviate their job insecurity:

**Hypothesis** **5** **(H5).***Job insecurity has a significant impact on family motives*.

**Hypothesis** **6** **(H6).***Family motivation has a significant impact on unethical workplace behavior*.

**Hypothesis** **7** **(H7).***Family motivation mediates the impact of job insecurity on unethical workplace behavior*.

## 3. Methodological Approach

### 3.1. Development of Study Measures and Instrument

The formulation of the scales in this study was based on an extensive survey of previously employed theoretical items in the literature. This survey creates four factors, each with their own set of items, which are then revised to fit the context of the hospitality industry (Hotels and travel agents). The operationalization of the study variables is depicted in Table 1.

We use the fundamental processes proposed by Maddox [56] and Churchill [57] to create 21 variables (questions) on a standard five-point Likert scale, where strongly disagree is 1 and strongly agree is 5. The items measuring job insecurity were created with a multi-item scale developed by Hellgren et al. [30] and employed by Elshaer and Azazz [3] in the tourism industry. Family financial pressures were operationalized by three variables based on the work of Conger et al. [58] and employed by Elshaer and Azazz [3] in the hotel industry. We adopted the five items of family motivation of Menges et al. [47]; a sample item is “It is important for me to do good for my family”. Finally, unethical workplace behavior in the name of the family was measured by seven items derived from Liu et al. [25]; a sample item is “I took advantage of my position in the company to make things more convenient for my family”.

The questionnaire was formerly in English; then, the back-translation approach was adopted [59]. Three qualified academics (obtained Ph.D. degrees from UK) translated the research questionnaire from its original language (English) to respondents’ language (Arabic). Furthermore, another three academics conducted the back translation from respondents’ language (Arabic) to English. The resulted revealed that the two versions were the same and consistent with no differences. Using extensive pre-testing and piloting stages, the research instrument was internally validated with input from six academics and fifteen employees in the field of the hospitality industry. The pilot participants indicated the measures’ high consistency and face and content validity. The final instrument was administered to 1000 employees working in five-star hotels and travel agents in greater Cairo, Egypt.

### 3.2. Process of Collecting Data

The study data were gathered in a three-waves process from 30 five-star hotels and 35 travel agents classified as category A in greater Cairo, Egypt. It was decided to use the three-wave approach, with at least a one-month period between each wave, in order to reduce the probability of common method variance (CMV) [60]. Participants in the first wave (W1) provided information about their demographics as well as their perceptions of job insecurity. After one month (W2), participants who had completed the first wave were surveyed regarding unethical workplace behavior in the name of the family. After one more month (W3), those who completed both the W1 and W2 surveys answered questions about family financial stress and family motivation. To ensure that the data collected in all three waves came from the same respondents, a coding system was used. The participants were chosen with the help of the hotels/travel agencies mangers who agreed to let us use their staff lists for scientific purpose. A total of 1000 non-managerial employees were randomly chosen to participate in the three waves (W1, W2, and W3) of the survey process; replies were obtained from 890, 800, and 770 people, respectively, demonstrating response rates of 89%, 80%, and 77%, respectively.

## 4. Results

### 4.1. Descriptive Statistics

Of the 770 who completed the questionnaires in the final third wave, 70% were male. Ten percent of those who responded in the final wave were between the ages of 18 and 23, 15% were 24–29 years old, 25% were 30–35 years old, 30% were 36–41 years old, and 20% were above 42 years old. In terms of education, 50% of the participants achieved high school level, 30% obtained a college degree, and 20% had a bachelor’s level or above. Regarding work experience, 30% of participants had experience for one year or less, 40% had work experience of two to three years, while participants who had work experience for more than five years accounted for 15% of the total targeted participants.

The variables mean ranged from 3.46 to 4.13, while the items’ standard deviation (S.D.) scores were from 0.649 to 1.293, suggesting that the study data are more distributed and less gathered around its mean value [61]. The skewness and kurtosis scores are not exceeding −2 or +2, suggesting that the study data have satisfactory normal distribution [62]. Furthermore, as depicted in Table 1, variance inflation factors (VIF) scores for all items were below 0.4, which confirm that multicollinearity is not an issue in our study [63].

### 4.2. Measurement Model Assessment

For construct reliability and validity, we used confirmatory factor analysis (CFA) to combine all dependent and independent unobserved latent variables into a single CFA model that showed a satisfactory model fit: model fit: χ^2^ (183, *n* = 770) = 489.891, *p* < 0.001, normed χ^2^ = 2.677, RMSEA = 0.031, CFI = 0.977, TLI = 0.978, NFI = 0.977), as depicted in Table 2 [61,64]. We examined the composite reliability and discriminant validity of our constructs using the estimates from this model [65]. Measuring the construct validity was completed by examining convergent and discriminant validity for each construct. To test the convergent validity of the factors, Table 2 shows that all factor loadings for our constructs’ items are statistically significant at the 0.001 level and exceed the minimum criterion of 0.5. Second, the average variance extracted (AVE) for all research constructs is greater than 0.5. Finally, the construct reliability (CR) scores for all the employed four factors—job insecurity (0.959), family financial pressure (0.939), family motivation (0.979), and unethical workplace behavior in the name of the family (0.977)—exceeded the recommended 0.70 cut-off point. Thus, as Anderson and Gerbing [64] and Hair et al. [62] recommend, our CFA output results revealed that all research constructs have a high satisfactory level of convergent validity.

The discriminant validity of constructs was assessed using Cronbach alpha values, correlation matrixes, and the square root of AVEs, as recommended by Fornell and Larcker [65]. Table 2 showed the correlation matrix, composite Cronbach alphas, and AVE values of the research four factors. As displayed in Table 2, bold diagonal values (the square root of AVEs) are larger than off-diagonal values (the correlations between those factors), which support a satisfactory discriminant validity for research factors as advocated by Fornell and Larcker [65]. Finally, the AVE values for all the four factors surpass the maximum shared values (MSV), further supporting a satisfactory level of discriminant validity. Overall, our measurement model demonstrates satisfactory levels of composite reliability and discriminant validity, according to the previous findings.

### 4.3. Structural Model Assessment

Following the establishing of confidence in the adequacy of the employed measures, we conducted structural equation modeling (SEM) to test the impact of job insecurity on unethical workplace behavior in the name of the family via family financial pressure and family motivation. The evaluation of the hypothesized model is confirmed through two main criteria: (1) the overall model goodness of fit (GoF) using the recommended indices such as x^2^/df, TLI, CFI, RMR and RMSEA and the statistical significance level for the models’ hypotheses. As shown in Table 3, the GoF measures for the structural model yielded satisfactory results. Additionally, the results of the anticipated model are illustrated in Figure 2 and Table 3.

The interrelationship in the suggested model contains seven justified hypotheses that investigate interactions among the research latent variables. The SEM analysis revealed that all seven hypotheses are significant at a statistical *p*-value less than 0.05. Hypothesis 1 (H1) investigated the direct effect of job insecurity on unethical workplace behavior in the name of the family, which was supported (*T*-value = 3.759, *p* < 0.01) with a path coefficient of 0.21, demonstrating that the two variables have a positive direct relationship. Likewise, the SEM analysis revealed that the job insecurity significantly and positively affects family financial pressure (H2) (*T*-value = 12.479, *p* < 0.001) with a path coefficient of 0.39, thus supporting hypothesis number 2 (H2). Additionally, in line with our proposition, the effect of family financial pressure on unethical workplace behavior in the name of the family was found to be significant and positive (*T*-value = 13.018, *p* < 0.001) with a correlation coefficient of 0.45, therefore supporting hypothesis number 3 (H3). Furthermore, as proposed, hypothesis number five tested the impact of job insecurity on family motivation, and the analysis gave signals of a positive and significant (*T*-value = 10.444, *p* < 0.001) association between the two factors with a correlation coefficient of 0.37, thus supporting hypothesis number 5 (H5). Finally, as proposed in hypothesis number 6 (H6), the impact of family motivation on unethical workplace behavior in the name of the family was found to be significant and positive (*T*-value = 15.702, *p* < 0.001, coefficient = 0.49); thus, hypothesis number 6 (H6) was confirmed.

To test hypotheses 4 and 7, suggestions introduced by [62,66] were adopted to evaluate the mediation impact of family financial pressure and family motivation in the relationship between job insecurity and unethical workplace behavior in the name of the family. Specifically, Zhao et al. [66] declared that for “direct-only nonmedication effects”, only direct path coefficients should be observed, and that only direct path coefficients should be observed with a significant p-value, while all indirect relationships should not be statistically significant. For “complementary mediation”, all direct and indirect correlations should be significant *p*-value with the same sign. Finally, “competitive mediation” is supported when all relationships (direct and indirect) are significant but with opposing signs. As pictured in Figure 2, all the tested paths are significant with the same positive sign, as depicted in Table 4. Specifically, the direct relationship between job insecurity and unethical workplace behavior in the name of the family is significant and positive (β = 0.21, *t*-value = 3.759, *p* < 0.001), and job insecurity directly, positively, and significantly, affects family financial pressure (β = 0.39, *t*-value = 12.479, *p* < 0.001) and family motivation (β = 0.37, *t*-value = 10.444, *p* < 0.001). In the same vein, family financial pressure was found to have a direct, positive, and significant relationship with unethical workplace behavior in the name of the family (β = 0.45, t-value = 13.018, *p* < 0.001). Similarly, family motivation was found to have a direct, positive, and significant relationship with unethical workplace behavior in the name of the family (β = 0.49, t-value = 15.702, *p* < 0.001). These results supported the complementary mediation of family financial pressure and family motivation in the relationship between job insecurity and unethical workplace behavior in the name of the family, therefore supporting hypotheses 4 and 7. Furthermore, specific indirect estimates from job insecurity to unethical workplace behavior through family financial pressure was calculated from the SEM Amos output (as shown in Table 4) to detect mediation, in which the lower (0.210) and the upper value (0.393) generated a significant (*p* > 001) standardized indirect estimates of 0.303. Similarly, the specific indirect estimate from job insecurity to unethical workplace behavior through family motivation was lower (0.216), and the upper value (0.389) created a significant (*p* > 001) standardized indirect estimate of 0.326. The previous results further support H4 and H7.

Finally, the standardized indirect path coefficient and total effects may also be reviewed in the SEM output to detect mediation impacts [62]. The standardized indirect path coefficients from the job insecurity to unethical workplace behavior in the name of the family via the mediating role of family financial pressure and family motivation increase the direct impact from 0.21 (β = 0.21, *p* < 0.01) to a total impact of 0.65 (β = 0.65, *p* < 0.001). This suggests that unethical workplace behavior in the name of the family increased by 44% via the mediating role of family financial pressure and family motivation. Furthermore, the proposed structural model showed a high level of explanatory power (R^2^), explaining 52% of the variation in unethical workplace behavior in the name of the family (Table 3).

## 5. Discussion and Implications

The outbreak of COVID-19 has created an opportunity to better understand the relationship between unethical behavior and perceived risk of job insecurity in tourism organizations. Drawing to theories of conservation of resources, social cognitive and behavioral ethics (i.e., the self-maintenance and bounded ethicality theories), this study examined family motivation and financial pressure as mediators of the relationship between job insecurity and unethical organizational behaviors. The results provide insights into the reasons behind the increased UOB during the COVID-19 pandemic [2,3,8,9], and they would add to researchers and practitioners understanding of how to address ethical challenges in their organizations.

A survey was created and distributed to 770 employees of five-star hotels and Category A travel agencies. To evaluate the proposed structural model as well as the measurement model, two main data analysis methods were used. SEM was used to evaluate the proposed structural model, and CFA was used to evaluate the measurement model’s convergent and discriminant validity, respectively. The findings revealed that the measurement model exhibited good convergence and discriminant validity, and that the proposed structural model accurately represented the data. A total of seven hypotheses were proposed and evaluated. The results revealed that job insecurity has a direct impact on unethical workplace behavior in the name of family as well as an indirect impact through financial pressure and motivation imposed by the extended family. The indirect effect increases the total effect of job insecurity on unethical organization behavior in the name of the family by 44%, providing evidence that both family financial pressure and family motivation play a role in mediating the relationship between job insecurity and unethical organization behavior in the name of the family. All endogenous variables combined account for 52% of the variance in unethical organizational behavior to benefit family, according to the findings.

These results extend the prior results of Lawrence and Kacmar [38], who found emotional exhaustion as a mediator in the relationship between job insecurity and unethical behavior. This is by highlighting the role of emotional exhaustion that derived from family financial pressures and motivation. Complementing prior research, family financial pressures [25,46] and motivations [9,55] were found to reduce self-regulatory resources and motivate unethical behaviors in the name of the family. The results indicated that participants who suffered high job instability in their workplace experienced higher levels of family pressures and were more prone to practice UOB to benefit their relatives. In support, Zhang et al. [46] articulated that when employees encounter substantial financial difficulties in their families, they are more likely to focus on obtaining financial benefits from their employer. In addition, this study provides empirical support for the argument of Hillebrandt and Barclay [4] that anxiety elicited by COVID-19 may focus employees’ attention on their self-related interest and encourage cheating and other unethical behaviors.

### 5.1. Implications to Theory

The result of this study has a three-fold contribution to theory. First, it extends the discussion of previous studies on conditions that shape unethical practices in workplaces. Most previous studies have identified that perceived job insecurity results in UOB that are either for benefiting the organization or self-serving. While this study adds empirical evidence on the important effect of family-related pressures, financial or motivational, on UOB, with particular attention of the influence of these factors when accompanied with job insecurity. Second, this study tries to answer the calls of prior researchers [3,4,22,25,66] to further examine the influence of environmental factors (e.g., COVID-19 outbreak) on UOB. The results add to explaining the reasons of the widespread unethical behaviors in organizations encountered during the COVID-19 pandemic. In particular, the current study adds to the understanding of the psychological process that employees may go through in making unethical decisions when faced by job insecurity. It is also the first study to discuss the mediating role of family motivation and financial pressure in the relationship between perceived job insecurity and UOB.

Third, this study answers calls to examine the employee’s family as a source of motivation to possible UOB [25,46]. Despite previous warnings of its possible strong influence on unethical workplace behavior [46], family motivation has received little empirical and theoretical attention. Instead, most studies have regarded family motivation as a driver of work effectiveness (e.g., [47]). Therefore, by demonstrating the role of family motivation in triggering unethical workplace behavior, this study extends research knowledge about possible psychological aspects that fuel UOB. At the same time, it gives insights to the possible dual role of family motivation as a source of desirable (work effectiveness, for more information see [47]) and undesirable organizational actions (UOB).

### 5.2. Implications to Practice

The tourism industry was the most affected during the COVID-19 pandemic. The large seen shutdowns and layoffs during the pandemic and their effect on employee’s psychology and behavior would have a long-lasting difficult influence on tourism workplaces and hospitality industry if not well understood and dealt with by decision-makers. Although the influence of COVID-19 on job loss and job insecurity has received much attention in recent tourism and hospitality research, a more comprehensive picture of the detrimental effect of job stress and anxiety on employee’s behavior still needs to be explored to better inform tourism decision-makers [9,23]. In this regard, this study contributes to the understanding of why the COVID-19 pandemic has induced UOB in the tourism industry and highlights the role of job insecurity and family pressures and motivation in increasing UOB in the name of the family. This would inform tourism decision-makers and managers toward preventing unethical behavior and the possible damaging consequences on financial performance and reputation. Decision-makers should strive to advocate social messages and enact policies that reduce employees’ perceived job insecurity, since this is a gateway to UOB.

In addition, based on the concept of bounded ethicality, the current study asserted that ambiguous circumstances (e.g., job insecurity) may veil the ethical aspects of employee’s decisions, as people lean to self-justification of their immoral actions in the workplace [17,18,67]. Therefore, pressures of job loss may drive even honest employees to engage in minor dishonesty and simultaneously stay ignorant (ethically blind) of the ethical repercussions of their acts [18,68]. Thus, with this in mind, organizational managers need to carefully observe the actions of all employees and regularly provide moral reminders, which can be a useful tool to reduce or prevent UOB. Organizations also should assess the ethical development of employees, promote moral actions and punish for unethical behavior.

## 6. Conclusions, Limitations, and Further Research

The prevalent and costly unethical practices in workplaces and their different types have attracted recent research attention, especially during the COVID-19 crises. This study investigated a neglected yet important form of UOB: unethical organizational practices in the name of the family. It proposed a model that may assist academics and practitioners in better understanding of how perceived job insecurity influences UOB through the mediating effect of family financial pressure and family motivation. The results revealed that perceived risk of job insecurity predicts employees’ engagement in UOB, while intentions of unethical behaviors increase by high family motivation and financial pressures.

However, this study has some limitations that offers opportunities for future research papers. The current results of the analyzed data showed that family motivation and financial pressure partially mediated the impact of job insecurity and unethical workplace behavior. To further our understanding on the relationship between job insecurity and UOB, future research papers can investigate more mediating variables (e.g., work intensification, trust in management, feeling of guilt, and job embeddedness) that can affect the relationship between job insecurity and unethical workplace behavior in the name of the family. Moreover, future studies should investigate possible boundary conditions such as moral identities and religious commitment, since previous studies, for example, revealed that moral identity undermines the strong influence of the self-control depletion on dishonesty and unethical practices [69].

Although this study ensured the confidentiality and anonymity of the questionnaire for the participants, the self-reported survey used in this study may encourage participants to biases their answers, since questions were about unethical actions to benefit the family. Thus, future studies can allow peer evaluation through colleagues or supervisors for more objectivity. In addition, future research can address the practices of decision makers to diminish UOB for family benefit and suggest methods that can be followed to control unethical practices. Furthermore, the collected data were cross-sectional; therefore, causal association between latent variables cannot be completely confirmed, and it is recommended for future investigations to collect longitudinal objective data or a different data source to validate the study model. Finally, a multigroup analysis approach can be conducted in future studies to validate and compare the results of the current study with data collected from different context (industry/country) [70]. Finally, it is important to highlight that the current study explores the relationship between family pressure and unethical workplace behavior under job insecurity. Accordingly, the UOB examined is not general but rather related to the benefit of the family. Future research would need to study general and other specific unethical behaviors that prevail in times of crisis (e.g., under job insecurity) and apply them to different contexts to understand how prevalent UOB is and how to mitigate its undesired consequences.

## Figures and Tables

**Figure 1 ijerph-19-06541-f001:**
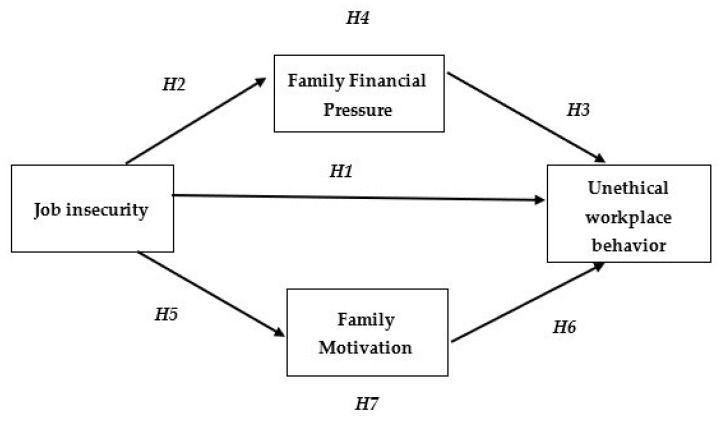
Research framework.

**Figure 2 ijerph-19-06541-f002:**
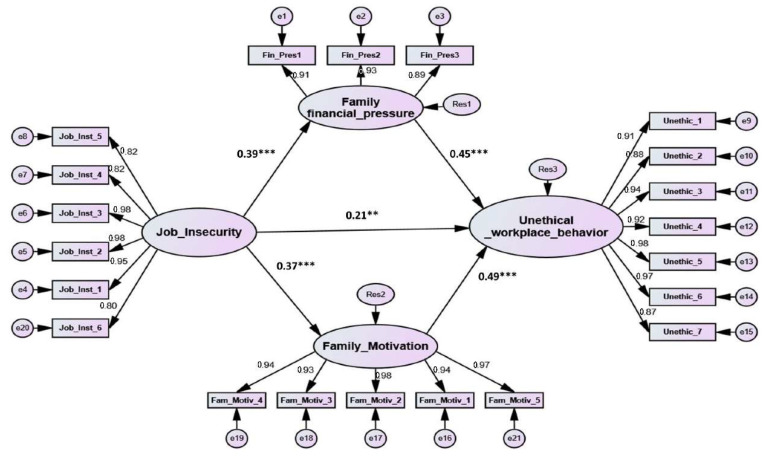
The structural model. ***: significant level below 0.001, **: significant level below 0.01.

**Table 1 ijerph-19-06541-t001:** Descriptive statistics.

Abbreviation	Items	M	S.D	Skewness	Kurtosis	VIF
**Job insecurity**—Job_Inst—Hellgren et al. [30]						
Job_Inst_1	“I am worried that I will have to leave my job before I would like to.”	3.51	1.1496	−0.412	−0.578	1.110
Job_Inst_2	“I worry about being able to keep my job.”	3.51	1.163	−0.399	−0.634	3.542
Job_Inst_3	“I am afraid I may lose my job shortly.”	3.54	1.125	−0.399	−0.561	3.529
Job_Inst_4	“I worry about getting less stimulating work tasks in the future.”	3.46	1.180	−0.424	−0.502	2.979
Job_Inst_5	“I worry about my future wage development.”	3.47	1.176	−0.427	−0.487	2.133
Job_Inst_6	“I feel worried about my career development in the organization.”	3.46	1.198	−0.471	−0.456	2.891
**Family financial pressure**—Fin_Pres—**Conger et al. [58]**						
Fin_Pres_1	“My family can hardly make ends meet.”	4.11	0.661	−1.288	1.601	1.589
Fin_Pres_2	“My family has difficulty paying its monthly bills.”	4.13	0.671	−1.306	1.404	1.137
Fin_Pres_3	“My family has little money left at the end of the month.”	4.10	0.649	−1.283	1.922	1.464
**Family Motivation**—Fam_Motiv—Menges et al. [47]						
Fam_Motiv_1	“I care about supporting my family.”	3.58	1.239	−0.412	−0.966	2.101
Fam_Motiv_2	“I want to help my family.	3.53	1.188	−0.339	−0.940	3.065
Fam_Motiv_3	“I want to have a positive impact on my family.”	3.61	1.168	−0.341	−0.977	1.290
Fam_Motiv_4	“It is important for me to do good for my family.”	3.48	1.208	−0.338	−0.959	2.141
Fam_Motiv_5	“My family benefits from my job.”	3.51	1.202	−0.359	−0.927	3.842
**Unethical workplace behavior in the name of the family**—Unethic—**Liu et al. [25]**						
Unethic_1	“To help my family, I took company assets/supplies home for family use.”	3.82	1.248	−1.023	0.062	3.196
Unethic_2	“To help my family, I submitted my family’s household receipts (e.g., gas) to my company for reimbursement.”	3.74	1.263	−0.937	−0.158	3.250
Unethic_3	“I took my family members to work to enjoy company resources and benefits that were intended for employees.”	3.78	1.241	−0.987	0.033	3.556
Unethic_4	“I took advantage of my position in the company to make things more convenient for my family.”	3.77	1.252	−0.997	0.004	3.740
Unethic_5	“I helped my family member get a job in my organization, even though I knew the family member was not qualified.”	3.73	1.270	−0.923	−0.160	3.477
Unethic_6	“I disclosed confidential company information to my family members so that they can have advantages/benefits.”	3.72	1.293	−0.934	−0.203	2.471
Unethic_7	“To help my family, I spent work resources to deal with family-related issues when at work.”	3.70	1.293	−0.892	−0.290	3.166

**Table 2 ijerph-19-06541-t002:** CFA Discriminant and Convergent Validity.

Factors and Items	Loading	CR	AVE	MSV	1	2	3	4
**1—Job Insecurity (*a* = 0.905)**		0.959	0.799	0.33	**0.894**			
Job_Inst_1	0.948							
Job_Inst_2	0.976							
Job_Inst_3	0.979							
Job_Inst_4	0.823							
Job_Inst_5	0.817							
Job_Inst_6	0.800							
**2—Family Financial Pressure (*a* = 0.917)**		0.939	0.837	0.33	0.182	**0.915**		
Fin_Pres1	0.914							
Fin_Pres2	0.935							
Fin_Pres3	0.895							
**3—Family Motivation (*a* = 0.902)**		0.979	0.904	0.13	0.47	0.053	**0.951**	
Fam_Motiv_1	0.939							
Fam_Motiv_2	0.977							
Fam_Motiv_3	0.931							
Fam_Motiv_4	0.937							
Fam_Motiv_5	0.970							
**4—Unethical Workplace Behavior (*a* = 0.918)**		0.977	0.857	0.16	0.125	0.11	0.43	**0.926**
Unethic_1	0.913							
Unethic_2	0.884							
Unethic_3	0.940							
Unethic_4	0.923							
Unethic_5	0.976							
Unethic_6	0.971							
Unethic_7	0.868							

**Model fit: (χ^2^ (183, *n* = 770) = 489.891, *p* < 0.001, normed χ^2^ = 2.677, RMSEA = 0.031, SRMR = 0.039, CFI = 0.977, TLI = 0.978, NFI = 0.977, PCFI = 0.809 and PNFI = 0.810)**. CR: composite reliability; AVE: average variance extracted; MSV: maximum shared value. Diagonal values: the square root of AVE for each dimension. Below diagonal values: intercorrelation between dimensions.

**Table 3 ijerph-19-06541-t003:** Result of structural model.

	Hypotheses	Beta(β)	C-R(*T*-Value)	R^2^	Hypotheses Results
**H1**	**Job insecurity**	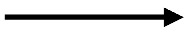	**Unethical workplace behavior**	**0.21 ****	**3.759**		Supported
**H2**	**Job insecurity**	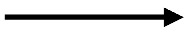	**Family financial pressure**	**0.39 *****	**12.479**		Supported
**H3**	**Family financial pressure**	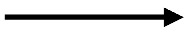	**Unethical workplace behavior**	**0.45 *****	**13.018**		Supported
**H4**	**Job insecurity 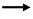 Family financial pressure 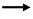 Unethical workplace behavior**	**Path 1: β = 0.39 ***;** ** *t* ** **-value = 12.479** **Path 2: β = 0.45 ***;** ** *t* ** **-value = 13.018**		Supported
**H5**	**Job insecurity**	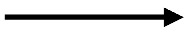	**Family motivation**	**0.37 *****	**10.444**		Supported
**H6**	**Family motivation**	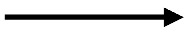	**Unethical workplace behavior**	**0.49 *****	**15.702**		Supported
**H7**	**Job insecurity 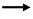 Family motivation 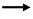 Unethical workplace behavior**	**Path 1: β = 0.37 ***;** ** *t* ** **-value = 10.444** **Path 2: β = 0.49 ***;** ** *t* ** **-value = 15.702**		Supported
**Unethical workplace behavior**					**0.52**	

**Model fit: (χ^2^ (184, *n* = 770) = 677.488, *p* < 0.001, normed χ^2^ = 3.682, RMSEA = 0.043, SRMR = 0.049, CFI = 0.974, TLI = 0.975, NFI = 0.967, PCFI = 0.709 and PNFI = 0.710). *** *p* < 0.001, ** *p* < 0.01.**

**Table 4 ijerph-19-06541-t004:** Calculating mediation effects from Amos output.

Indirect Path	Unstandardized Estimate	Lower	Upper	*p*-Value	Standardized Estimate
Job Insecurity → Family financial pressure → Unethical workplace behavior	0.341	0.210	0.393	0.001	0.303 ***
Job Insecurity → Family Motivation → Unethical workplace behavior	0.334	0.216	0.389	0.001	0.326 ***

*** *p* < 0.001.

## Data Availability

Data are available upon request from researchers who meet the eligibility criteria. Kindly contact the first author privately through e-mail.

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
