# Peer review of "An Unethical Organizational Behavior for the Sake of the Family: Perceived Risk of Job Insecurity, Family Motivation and Financial Pressures"

_ijerph, 2022, doi:10.3390/ijerph19116541_

Round 1
Reviewer 1 Report
The authors submitted article entitled " An unethical organizational behavior for the sake of the family: perceived risk of job insecurity, family motivation and financial pressures" is not a appropriate research problem at this stage of COVID-19 pandemic situation. Especially, this research conducted on tourism employees and organization which is even more panic industry during COVID-19 pandemic. It is difficult to understand the ethical or unethical behavior of employees and organization due to uncontrolled situation.
Therefore, I am not supporting this article to publish in IJERPH journal.
Author Response
Dear Reviewer1,
Thank you for giving us the opportunity to submit a revised draft of our manuscript titled “An unethical organizational behavior for the sake of the family: perceived risk of job insecurity, family motivation and financial pressures.” to IJERPH journal. We appreciate the time and effort that you have dedicated to providing your valuable feedback on our manuscript. We are grateful to the reviewers for his insightful comments on our paper. We have been able to incorporate changes to reflect the suggestions provided in our revised manuscript. We have colored the changes within the manuscript in red.
attached is a point-by-point response to the reviewers’ comments and concerns.

Reviewer 2 Report
The manuscript “An unethical organizational behavior for the sake of the family: perceived risk of job insecurity, family motivation and financial pressures” is timely and very interesting. However, I have some concerns that should be addressed before it can be considered for publication in the International Journal of Environmental Research and Public Health.
My main concerns about the manuscript are outlined below:
- I found the introduction not developed enough, and the theoretical framework which explains why job insecurity leads to unethical behavior is not developed. First, the authors claim that unethical behavior reduces the threat to the employee’s career. But they do not explain why, and do not refer to the possibility that behaving unethically in the workplace might hurt work safety. Second, the authors do not discuss dated theories in behavioral ethics (e.g., the self-maintenance theory and bounded ethicality) to explain why family pressure and motivation can lead to unethical behavior (and especially, they need to refer to known factors which promote unethical behavior, e.g., justifications, and link them to the context of family pressure and the workplace).
- Relatedly, the authors need to explain why financial pressure from the family side motivates people to reduce the pressure (p. 3, lines 129-132).
- The authors present a mediation model in Figure 1, but discuss correlations in the introduction and hypotheses. The authors need to better exaplin why family motivation and financial pressure mediate the effect of job insecurity on unethical workplace behavior.
- The current work examines the link between family pressure and unethical workplace behavior under job insecurity. But the unethical behavior is not general but rather related to the benefit of the family. This focus does not allow the authors to examine if unethical behavior, in general, prevails in times of crisis (e.g., under job insecurity), or to understand how prevalent this behavior actually is. The authors should refer to this limitation.
- Since the authors only use self-reports, the data might suffer from social desirability. That is, since the authors ask about unethical things that people will do for their family, it might encourage participnats to report doing things for the family. Obviously, there is nothing to do now about this limitation. But the authors should refer to it and try to explain why their results are valid even under this.
- Drawing conclusions about a mediating mechanism requires a strong theoretical framework, which this study lacks. Moreover, the researchers should establish a clear temporal sequence to show mediation – the mediating variable should happen before the DV. In the current research, the authors measure the DV before the mediating variable. Why? And how can they draw casual relationships based on this?
- Based on these comments and concerns, the authors should revise the discussion and conclusion, attenuate some of the claims, and better discuss limitations. Currently, they focus in the limitations section only on things they didn’t do (e.g., testing other mediators), and not on discussing the issues with the method they used.
Minor comments:
- 1, l. 32. Please explain what WTTC stands for
- 2, l. 64 – you should explain what UOB stands for here, and not on l. 93 (same page)
- 3, l. 101, delete the “:”
- The sentence is p. 3, l. 102 is not clear. Please rephrase
- Figure 1 does not have a heading, and Figure 2 is labeled Fig. 1
- Delete the “from” at the end of l. 216 on p. 5
- 5, lines 221-223 – these sentences are phrased strangely. Please rephrase
- 11, l. 315 – capitalize “in”
- Due to the fact that ego depletion is not replicated, you might want to reconsider using this theory in your intro (references 27 and 28)
I hope the authors will find these comments useful as they continue with this interesting project
Author Response
Dear Reviewer, 2
Thank you for giving us the opportunity to submit a revised draft of our manuscript titled “An unethical organizational behavior for the sake of the family: perceived risk of job insecurity, family motivation and financial pressures.” to IJERPH journal. We appreciate the time and effort that you have dedicated to providing your valuable feedback on our manuscript. We are grateful to the reviewers for his insightful comments on our paper. We have been able to incorporate changes to reflect the suggestions provided in our revised manuscript. We have colored the changes within the manuscript in red.
Attached is a point-by-point response to the reviewers’ comments and concerns.

Round 2
Reviewer 1 Report
Dear author thank you for giving me detailed explanation for my previous comments. After understanding the importance of this type of research design I would like to accept this article to publish in IJERPH.